# Satisficing Regret Minimization in Bandits

**Qing Feng**
Cornell University
qf48@cornell.edu

**Tianyi Ma**
Shanghai Jiao Tong University
sean_ma@sjtu.edu.cn

**Ruihao Zhu**
Cornell University
ruihao.zhu@cornell.edu

## Abstract

Motivated by the concept of *satisficing* in decision-making, we consider the problem of satisficing regret minimization in bandit optimization. In this setting, the learner aims at selecting satisficing arms (arms with mean reward exceeding a certain threshold value) as frequently as possible. The performance is measured by *satisficing regret*, which is the cumulative deficit of the chosen arm's mean reward compared to the threshold. We propose SELECT, a general algorithmic template for Satisficing REgret Minimization via SampLing and LowEr Confidence bound Testing, that attains constant satisficing regret for a wide variety of bandit optimization problems in the *realizable* case (*i.e.*, a satisficing arm exists). Specifically, given a class of bandit optimization problems and a corresponding learning oracle with sub-linear (standard) regret upper bound, SELECT iteratively makes use of the oracle to identify a potential satisficing arm with low regret. Then, it collects data samples from this arm, and continuously compares the LCB of the identified arm's mean reward against the threshold value to determine if it is a satisficing arm. As a complement, SELECT also enjoys the same (standard) regret guarantee as the oracle in the non-realizable case. Finally, we conduct numerical experiments to validate the performance of SELECT for several popular bandit optimization settings.

## 1 Introduction

Multi-armed bandit (MAB) is a classic framework for decision-making under uncertainty. In MAB, a learner is given a set of arms with initially unknown mean reward. She repeatedly picks an arm to collect its noisy reward and her goal is to maximize the cumulative reward. The performance is usually measured by the notion of *regret*, which is the difference between the maximum possible total mean reward and the total mean reward collected by the learner. With its elegant form, MAB has found numerous applications in recommendation systems, ads display, and beyond, and many (near) optimal algorithms have been developed for various bandit optimization settings (Lattimore & Szepesvári, 2020).

While optimal decision is desired in certain high-stakes situations, it is perhaps not surprising that what we often need is just a good enough or *satisficing* alternative (Lufkin, 2021). Coined by Nobel laureate Herbert Simon (Simon, 1956), the term "satisficing" refers to the decision-making strategy that settles for an acceptable solution rather than exhaustively hunting for the best possible.

The concept of satisficing finds its place in many real world decision-making under uncertainty settings (Caplin et al., 2011). For example, in inventory management, studies show that product stockouts can result in customer disengagement and impair a firm's profitability (Fitzsimons, 2000; Jing & Lewis, 2011). As such, retailers usually replenish their products periodically to improve service level (*i.e.*, probability of no stockout for any product). Notably, due to various constraints (*e.g.*, capacity and inventory cost), instead of blindly maximizing service level, retailers typically set a target service level (*e.g.*, the ideal service level in retailing is $90\% \sim 95\%$ (Levy et al., 2023)). Consequently, in face of the unknown demand distributions, one can cast the problem into a MAB setting with a *different* objective. In this setting, a retailer starts without complete knowledge of the demand distributions and proceeds by trying various stock level combinations (*i.e.*, arms) of the products over time. Here, the objective is to hit the target service level as frequently as possible rather than maximizing the service level every time. As another example, in fashion product design, designers usually need to design a sequence of products for a group of customers with initially

unknown preferences. Instead of demanding for the best design every time, designers only need to pick a design that can hopefully meet the expectation of a certain portion of the customers for most of the time (Mabey, 2022; Whitenton, 2014).

The above examples give rise to the study of satisficing bandits (Tamatsukuri & Takahashi, 2019; Michel et al., 2023) where the objective is to hit a satisficing arm (*i.e.*, an arm whose mean reward exceeds a certain threshold value) as frequently as possible. The corresponding performance measure is *satisficing regret*, which is the cumulative deficit of the chosen arm's mean reward compared to the threshold. In particular, Michel et al. (2023) develops SAT-UCB, an algorithm for finite-armed satisficing bandits: In the *realizable case* (*i.e.*, a satisficing arm exists), the algorithm attains a constant satisficing regret, *i.e.*, independent of the length of the time horizon; Otherwise, in the non-realizable case (*i.e.*, the threshold value is out of reach), it preserves the optimal logarithmic (standard) regret.

Despite these, existing algorithms usually impose that there is a clear separation between the threshold value and the largest mean reward among all *non-satisficing arms*, which is referred to as the *satisficing gap*. Moreover, the satisficing regret bounds usually scale *inverse proportionally* to the satisficing gap. On the other hand, it is evident that in many bandit optimization problems (*e.g.*, Lipschitz bandits and concave bandits), the satisficing gap can simply be 0, making the bounds vacuous. Therefore, it is not immediately clear if SAT-UCB (Michel et al., 2023) or other prior results can go beyond finite-armed bandits and if constant satisficing regret is possible for bandit optimization in general.

**Main Contributions:** In this work, we propose SELECT, a novel algorithmic template for Satisficing REgret Minimization via SampLing and LowEr Confidence Bound Testing. More specifically:

- We describe the design of SELECT in Section 3. For a given bandit optimization problem class and an associated learning oracle with sub-linear (but not necessarily optimal) standard regret guarantee, SELECT runs in rounds with the help of the oracle. At the beginning of a round, SELECT first runs the oracle for a number of time steps and randomly samples an arm from its trajectory as a candidate satisficing arm. Then, it conducts forced sampling by pulling this arm for a pre-specified number of times. Finally, it continues to pull this arm and starts to monitor the resulted lower confidence bound (LCB) of its mean reward. SELECT terminates the current round and starts the next once the LCB falls below the threshold value;

- In Section 4, we establish that in the realizable case (*i.e.*, when a satisficing arm exists), SELECT is able to achieve a constant satisficing regret. Notably, the satisficing regret bound has no dependence on the satisficing gap (see the forthcoming Remark 2 for a detailed discussion). Instead, it scales inverse proportionally to the *exceeding gap*, which measures the difference between the optimal reward and the threshold value, and is positive in general; Otherwise (*i.e.*, when no satisficing arm exists), SELECT enjoys the same standard regret bound as the learning oracle.

- In Section 5, we instantiate SELECT to finite-armed bandits, concave bandits, and Lipschitz bandits, and demonstrate the corresponding satisficing and standard regret bounds. We also provide some discussions on the respective lower bounds;

- Finally, in Section 6 we conduct numerical experiments to demonstrate the performance of SELECT in fnite-armed bandits, concave bandits, and Lipschitz bandits. We use SAT-UCB, SAT-UCB+ (a heuristic with no regret guarantee proposed by Michel et al. (2023)), and the respective learning oracle as the benchmarks. Our results reveal that in the realizable case, SELECT does achieve constant satisficing regret. Moreover, both its satisficing regrets (in the realizable cases) and standard regrets (in the non-realizable cases) either significantly outperform the benchmarks' or are close to the best-performing ones'.

## 1.1 RELATED WORKS

Besides Michel et al. (2023), the most relevant works to ours are Bubeck et al. (2013); Garivier et al. (2019); Tamatsukuri & Takahashi (2019), all of which focus on finite-armed bandits. Bubeck et al. (2013) and Garivier et al. (2019) introduce constant regret algorithms with knowledge of the optimal reward. This can be viewed as a special case of satisficing where the threshold value is exactly the optimal mean reward. Hüyük & Tekin (2021) further extend the algorithm in Garivier et al. (2019)

to multi-objective multi-armed bandit settings. Tamatsukuri & Takahashi (2019) also establishes constant regret for finite-armed satisficing bandits when there exists exactly one satisficing arm. As a follow-up paper of Michel et al. (2023), Hajiabolhassan & Ortner (2023) considers satisficing in finite-state MDP.

Among others, Russo & Van Roy (2022) is one of the first to introduce the notion of satisficing regret, but they focus on a time-discounted setting. Furthermore, in Russo & Van Roy (2022), the learner is assumed to know the difference between the values of the threshold and the optimal reward.

More broadly, the concept of satisficing has also been studied by Reverdy & Leonard (2014); Reverdy et al. (2017); Abernethy et al. (2016) in bandit learning. Both Reverdy & Leonard (2014) and Abernethy et al. (2016) focus on maximizing the number of times that the selected arm's *realized reward* exceeds a threshold value. This is equivalent to a MAB problem that sets each arm's mean reward as its probability of generating a value that exceeds the threshold. In view of this, the problems investigated in Reverdy & Leonard (2014) and Abernethy et al. (2016) are closer to conventional MABs. They are also different from Reverdy et al. (2017); Michel et al. (2023) and ours, which use the mean reward of the selected arms to compute the satisficing regret. We note that in Reverdy et al. (2017), a more general Bayesian setting is studied. In this setting, the satisficing regret also takes the learner's belief that some arm is satisficing into account.

The notion of satisficing is also studied in the pure exploration setting (Locatelli et al., 2016; Mukherjee et al., 2017; Kano et al., 2019) as *thresholding bandits*. Their objective is to identify *all* arms with mean reward above a certain satisficing level with limited exploration budget. Unlike the satisficing regret we consider, the performance of their algorithm is measured by simple regret, *i.e.*, the expected number of misidentification made by the final output. Note that any pure exploration algorithm (Audibert & Bubeck, 2010) is able to identify the optimal arm or a satisficing arm with high probability after a number of steps. However, due to a small but positive error probability, subsequent exploitation of the identified arm will always incur linear satisficing regret, thus a simple explore-then-exploit approach is unable to obtain a constant satisficing regret bound. Similar observations have also been made in Michel et al. (2023).

## 2 PROBLEM FORMULATION

In this section, we present the setup of our problem. Consider a class of bandit optimization problem $(\mathcal{X}, \mathcal{R})$, where $\mathcal{X}$ is the arm set and $\mathcal{R} \subseteq \{r : \mathcal{X} \to \mathbb{R}\}$ is the class of admissible reward functions. The learner is given a set of feasible arms $\mathcal{X}$ and a class of admissible reward functions $\mathcal{R}$, pulling arm $X \in \mathcal{X}$ in a time step brings a random reward with mean $r(X)$. The learner knows the underlying reward function $r(\cdot)$ belongs to the function class $\mathcal{R}$, but the exact $r(\cdot)$ is initially unknown to the learner.

In each time step $t = 1, 2, \ldots, T$, the learner pulls an arm $X_t \in \mathcal{X}$, and then observes and receives a noisy reward $Y_t = r(X_t) + \epsilon_t$, where $\epsilon_t$ is a conditionally 1-subgaussian random variable, *i.e.*, $\mathbb{E}[\epsilon_t | X_1, \epsilon_1 \ldots, X_{t-1}, \epsilon_{t-1}, X_t] = 0$ and $\mathbb{E}[e^{\lambda \epsilon_t} | X_1, \epsilon_1 \ldots, X_{t-1}, \epsilon_{t-1}, X_t] \leq e^{\lambda^2/2}$ holds for all $\lambda \in \mathbb{R}$.

**Remark 1.** *The notion of problem class $(\mathcal{X}, \mathcal{R})$ captures most of the bandit optimization problems in the stationary setting. For example, if $\mathcal{X} = [K]$ and $\mathcal{R} = \{r : [K] \to \mathbb{R}\}$, then the problem class $(\mathcal{X}, \mathcal{R})$ corresponds to finite-armed bandits with $K$ arms; if $\mathcal{X}$ is any convex set in $\mathbb{R}^d$ and $\mathcal{R} = \{r : \mathcal{X} \to \mathbb{R}, r$ is concave and 1-Lipschitz$\}$, then the problem class $(\mathcal{X}, \mathcal{R})$ corresponds to concave bandits.*

In this work, we define two notions of regret. First, with the presence of satisficing level $S$, we consider the satisficing regret introduced in Reverdy et al. (2017); Michel et al. (2023), which is the expected cumulative deficit of the chosen arm's mean reward when compared to $S$, *i.e.*

$$\text{Regret}_S = \mathbb{E}\left[\sum_{t=1}^{T} \max\{S - r(X_t), 0\}\right].$$

Because it is possible that no arm achieves the satisficing level, we also follow the conventional bandit literature to define the (standard) regret, which measures the expected cumulative difference of

the mean reward between the optimal arm and the chosen arm's, *i.e.*, let $X^* = \arg\max_{X \in \mathcal{X}} r(X)$, the regret is defined as

$$\texttt{Regret} = T \cdot r(X^*) - \mathbb{E}\left[\sum_{t=1}^{T} r(X_t)\right].$$

We say that the problem is *realizable* if $r(X^*) \geq S$; Otherwise, it is *non-realizable*. We recall that an arm $X$ is satisficing if $r(X) \geq S$; Otherwise, it is non-satisficing. Existing works on similar settings (*e.g.*, Michel et al. 2023) usually derive satisficing regret bound that scales inverse proportional to the *satisficing gap*

$$\Delta_S = \min\{S - r(X) : r(X) < S\}.$$

However, for bandit optimization problems with large and even infinitely many arms, $\Delta_S$ can approach 0 quickly. To address this issue, we define the notion of *exceeding gap* that captures the difference between the optimal reward and $S$, *i.e.*,

$$\Delta_S^* = r(X^*) - S.$$

## 3  SELECT: SATISFICING REGRET MINIMIZATION VIA SAMPLING AND LOWER CONFIDENCE BOUND TESTING

In this section, we present our algorithmic template SELECT for general bandit problems.

When the arm space $\mathcal{X}$ is not too large, one can collect enough data samples for every arm to estimate its mean reward and gradually abandon any non-satisficing arms (see, *e.g.*, Garivier et al. 2019; Michel et al. 2023). In general, however, the arm space can be large and may even contain infinitely many arms (*e.g.*, continuous arm space). Therefore, it becomes impossible to identify all non-satisficing arms. We thus take an alternative approach: It repeatedly locates a potential satisficing arm with low (satisficing) regret. Then, it tests if this candidate arm is truly a satisficing arm or not. If not, it kicks off the next round of search.

To ensure fast detection of candidate satisficing arms with low regret for a given bandit problem class $(\mathcal{X}, \mathcal{R})$, SELECT makes use of a bandit algorithm for (standard) regret minimization on $(\mathcal{X}, \mathcal{R})$. To this end, we assume access to a blackbox learning oracle that achieves sub-linear standard regret for the bandit problem class $(\mathcal{X}, \mathcal{R})$.

**Condition 1.** *For problem class $(\mathcal{X}, \mathcal{R})$, there exists a sequence of learning algorithms ALG such that for any reward function $r \in \mathcal{R}$ and any time horizon $t \geq 2$, the regret of ALG(t) is bounded by*

$$\texttt{Regret} = \mathbb{E}\left[t \cdot r(X^*) - \sum_{s=1}^{t} r(X_s)\right] \leq C_1 t^\alpha \log(t)^\beta.$$

*where $C_1 \geq 1, 1/2 \leq \alpha < 1, \beta \geq 0$ are constants only dependent on the problem class and independent from the time horizon $t$.*

We remark that Condition 1 only asks for sub-linearity in standard regret upper bounds rather than optimality. Therefore, it is an extremely mild condition and can be easily satisfied by a wide range of bandit optimization problems and the corresponding bandit learning oracles, *e.g.*, finite-armed bandits, Lipschitz bandits, etc. We will demonstrate this in the forthcoming Section 5 and Appendix D of the full version (Feng et al., 2025).

With this, for a problem class $(\mathcal{X}, \mathcal{R})$ and its sub-linear regret learning algorithm ALG, SELECT runs in rounds. In round $i$, it takes the following steps (see Fig. 1 for a illustration):

**Step 1. Identify a Potential Satisficing Arm:** Let $\gamma_i = 2^{-i(1-\alpha)/\alpha}$, SELECT follows ALG($t_i$) for the first $t_i = \lceil \gamma_i^{-1/(1-\alpha)} \rceil = \lceil 2^{i/\alpha} \rceil$ steps of round $i$ and records its arm selections. Then, it samples an arm $\hat{X}_i$ uniformly random from the trajectory of ALG($t_i$) in this round. By virtue of ALG, we find an arm whose mean reward is at most $\tilde{O}(t_i^{\alpha-1}) = \tilde{O}(\gamma_i)$ below $r(X^*)$ in expectation, and only $\tilde{O}(t_i^\alpha)$ satisficing regret is incurred in this step in the realizable case. We note that in the realizable case, as $i$ increases, $\tilde{O}(\gamma_i)$ will gradually become smaller than $\Delta_S^*$, meaning that the sampled arm $\hat{X}_i$ is more likely to be a satisficing arm and enjoys no satisficing regret;

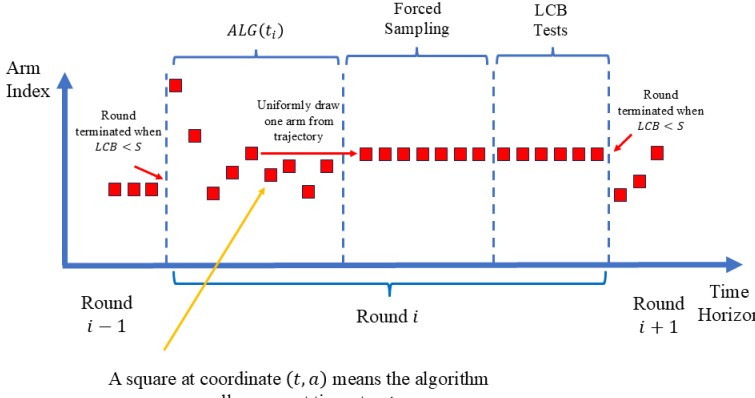

Figure 1: Illustration of a single round of SELECT

**Step 2. Forced Sampling on the Identified Arm:** To validate if $\hat{X}_i$ is a satisficing arm or not, we need to collect enough data from pulling $\hat{X}_i$ before performing any statistical test (see the forthcoming Remark 2 for the reasons). However, when the arm set $\mathcal{X}$ is large, $\text{ALG}(t_i)$ might only have pulled $\hat{X}_i$ for limited number of times. To circumvent this, we perform a forced sampling on $\hat{X}_i$ by pulling it for $T_i = \lceil \gamma_i^{-2} \rceil$ times;

**Step 3. Lower Confidence Bound Tests:** In this step, SELECT kicks off the LCB test as more data is collected from pulling $\hat{X}_i$. We use $k$ (initialized to 0) to denote the number of additional pieces of noisy reward data generated by pulling $\hat{X}_i$ in the current time step and $\hat{r}_i^{\text{tot}}$ to denote the running total reward collected from Step 2 and the current step so far. At the same time, it keeps comparing the LCB of $\hat{X}_i$'s mean reward, *i.e.*, $\hat{r}_i^{\text{tot}}/(T_i + k) - \sqrt{4\log(T_i + k)/(T_i + k)}$, against the threshold value $S$ after acquiring every piece of new data. SELECT terminates this round and enters the next whenever the LCB is less than $S$.

The pseudo-code of SELECT is presented in Algorithm 1.

**Remark 2.** *(Key Novelties of **SELECT**) Our algorithm brings together several key ingredients, including sampling from oracle trajectories (Step 1), forced sampling to collect data (Step 2), and the LCB test (Step 3). In what follows, we further comment on the subtleties of each step.*

*Step 1. Prior works on satisficing (Garivier et al. 2019; Michel et al. 2023) mainly use uniform exploration to find candidates of satisficing arms. This approach could incur major satisficing regret when the arm set is large or even possibly infinite, as conducting uniform exploration on a large arm set may result in pulling a large number of non-satisficing arms. To bypass this challenge, in Step 1, we search for candidates using a bandit learning oracle with sub-linear standard regret. Hence, we are able to find candidates of satisficing arms without incurring large regret.*

*Step 2. One challenge from using the LCB test in Step 3 is that it is a more conservative design compared to prior works. In fact, if we directly enter Step 3 without Step 2, the LCB of $r(\hat{X}_i)$ can easily fall below $S$ even if $r(\hat{X}_i) \geq S$. This is because we may not have pulled $\hat{X}_i$ for enough times in Step 1 and the corresponding confidence interval might be large. As a result, the lower confidence bound of $r(\hat{X}_i)$, which is the difference between empirical mean and confidence radius, can be significantly below $S$. This indicates, without the forced sampling in Step 2, major satisficing regret can be incurred due to frequent re-start of a new round. With the forced sampled data collected from Step 2, SELECT ensures that the width of the confidence interval is of order $\tilde{O}(T_i^{-1/2}) = \tilde{O}(\gamma_i)$ (i.e., same as $\mathbb{E}[r(X^*) - r(\hat{X}_i)]$). Consequently, whenever $\gamma_i$ shrinks to well below $\Delta_S^*$, SELECT gradually becomes less likely to terminate a round (see the forthcoming Proposition 2);*

*Step 3. In Step 3, SELECT compares the LCB of $\hat{X}_i$'s mean reward against $S$ to determine if $X_i$ is a satisficing arm. This deviates from prior works that use UCB (Garivier et al., 2019) or empirical mean (Michel et al., 2023). However, it turns out that this is essential in achieving a*

*constant satisfying regret that does not scale with $1/\Delta_S$, which can quickly explode in many cases. As we will see, once entering Step 3, `SELECT` can terminate a round within 1 additional time step in expectation when facing a non-satisfying arm. In contrast, if it follows prior works to use UCB or empirical mean instead, the number of time steps required (and hence, the satisfying regret) will unavoidably scale with $1/\Delta_S$ (see e.g., Theorem 9 of Garivier et al. 2019 or Theorem 1 of Michel et al. 2023).*

---

**Algorithm 1** Satisficing Regret Minimization via Sampling and Lower Confidence Bound Testing (`SELECT`)

---

**Input:** time horizon $T$, satisficing level $S$
$\gamma_i \leftarrow 2^{-i(1-\alpha)/\alpha}$ for all $i = 1, 2, 3 \ldots$
**for** round $i = 1, 2, \ldots$ **do**
    Set $t_i \leftarrow \lceil \gamma_i^{-1/(1-\alpha)} \rceil$
    Run `ALG`$(t_i)$, let $\{\bar{X}_s\}_{s \in [t_i]}$ be the trajectory of arm pulls
    Sample $R$ uniformly at random from $\{1, 2, \ldots, t_i\}$ and set $\hat{X}_i \leftarrow \bar{X}_R$
    Set $T_i \leftarrow \lceil \gamma_i^{-2} \rceil$, $k \leftarrow 0$
    Let $t$ be the current time step, pull arm $\hat{X}_i$ for the next $T_i$ time steps, and set $\hat{r}_i^{\text{tot}} \leftarrow \sum_{s=t}^{t+T_i-1} Y_s$
    **while** $LCB(\hat{X}_i) \geq S$ **do**
        Set $k \leftarrow k + 1$
        Pull arm $\hat{X}_i$ and observe $Y_t$, where $t$ is the current time step
        Update $\hat{r}_i^{\text{tot}} \leftarrow \hat{r}_i^{\text{tot}} + Y_t$
        Set

$$LCB(\hat{X}_i) \leftarrow \frac{\hat{r}_i^{\text{tot}}}{T_i + k} - \sqrt{\frac{4 \log(T_i + k)}{T_i + k}}$$

    **end while**
**end for**

---

## 4 REGRET ANALYSIS

In this section, we analyze the regret bounds of `SELECT`. We begin by providing an upper bound of satisficing regret in the realizable case, *i.e.*, $r(X^*) \geq S$.

**Theorem 1.** *If $r(X^*) \geq S$, then the satisficing regret of `SELECT` is bounded by*

$$\text{Regret}_S \leq \min \left\{ C_1^{\frac{1}{1-\alpha}} \left( \frac{1}{\Delta_S^*} \right)^{\frac{\alpha}{1-\alpha}} \cdot \text{polylog} \left( \frac{C_1}{\Delta_S^*} \right), C_1 T^\alpha \cdot \text{polylog}(T) \right\}. \tag{1}$$

We remark that, from the first term on the RHS of equation 1, `SELECT` can achieve a constant (w.r.t. $T$) satisficing regret in the realizable case. Moreover, even when $T$ is relatively small and the constant satisficing regret guarantee is loose compared to the oracle's regret bound, it is able to adapt to the oracle's performance.

*Proof Sketch.* A complete proof for Theorem 1 is provided in Appendix A of the full version (Feng et al., 2025). The proof relies on two critical results. In the first one, we show an upper bound on the satisficing regret of round $i$.

**Proposition 1.** *If $r(X^*) \geq S$, then the satisficing regret incurred in round $i$ is bounded by*

$$\frac{6C_1}{(1-\alpha)^\beta} \gamma_i^{-\alpha/(1-\alpha)} \log(2/\gamma_i)^\beta.$$

The proof of this proposition is provided in Appendix A.1 of the full version (Feng et al., 2025). We also show that once `SELECT` runs for enough rounds so that $\gamma_i = \tilde{O}(\Delta_S^*)$, it is unlikely to start a new round.

**Proposition 2.** *If $r(X^*) \geq S$, then for every $i$ that satisfies*

$$\frac{32C_1}{(1-\alpha)^\beta} \gamma_i \log(2/\gamma_i)^{\max\{\beta, 1/2\}} \leq \Delta_S^*,$$

*the probability that round $i$ ends within the time horizon $T$ (conditioned on the event that round $i$ is started within the time horizon) is bounded by $1/4$.*

The proof of this proposition is provided in Appendix A.2 of the full version (Feng et al., 2025). Since each round of SELECT runs independently, Proposition 2 indicates that the total number of rounds for SELECT can be upper-bounded by a shifted geometric random variable with success rate $3/4$. Combining the results of Proposition 1 and Proposition 2 we are able to establish the statement in Theorem 1. $\square$

We also show that, in the non-realizable case, SELECT enjoys the same regret bound as the oracle in Condition 1. The proof is provided in Appendix B of the full version (Feng et al., 2025).

**Theorem 2.** *If $r(X^*) < S$, the standard regret of SELECT is bounded by*

$$Regret \leq C_1 T^\alpha \cdot \text{polylog}(T).$$

With the above results, we establish that under a very general condition, SELECT does achieve constant satisficing regret in the realizable case without compromising the standard regret guarantee in the non-realizable case. Altogether, these empower SELECT the potential to become a general algorithm for decision-making under uncertainty.

## 5 EXAMPLES

In what follows, we instantiate SELECT to several popular bandit optimization settings, including finite-armed bandits, concave bandits, and Lipschitz bandits. Along the way, we showcase how our results enable constant satisficing regret for bandits with large and even infinite arm space, which was not achieved by existing algorithms. We remark that the examples given here are non-exhaustive and similar results can be derived for other bandit optimization settings such as linear bandits. In all three examples of this section, we assume the mean reward of each arm is in $[0, 1]$.

**1. Finite-Armed Bandits:** Consider a finite-armed bandit problem with $K$ arms, *i.e.*, $\mathcal{X} = [K]$. In this case, both the UCB algorithm (see, *e.g.*, Bubeck & Cesa-Bianchi 2012) and Thompson sampling (see, *e.g.*, Agrawal & Goyal 2017) achieve a regret bound of $O(\sqrt{KT \log(T)})$. Combining them with Theorems 1 and 2, we have the following corollary. At the end of this section, we also provide some discussions on the lower bounds of the satisficing regrets.

**Corollary 1.** *By using either the UCB algorithm or Thompson sampling as ALG, if $r(X^*) \geq S$, the satisficing regret of SELECT is bounded by*

$$Regret_S \leq \min\left\{ \frac{K}{\Delta_S^*} \cdot \text{polylog}\left(\frac{K}{\Delta_S^*}\right), \sqrt{KT} \cdot \text{polylog}(T) \right\};$$

*If $r(X^*) < S$, the standard regret of SELECT is bounded by $Regret \leq \sqrt{KT} \cdot \text{polylog}(T)$.*

**Remark 3** (Comparison with Garivier et al. 2019; Michel et al. 2023). *Garivier et al. (2019); Michel et al. (2023) also provide algorithms for satisficing regret for finite-armed bandits. While our satisficing regret bound is incomparable to Garivier et al. (2019), which attains a $O(K/\Delta_S)$ satisficing regret, we provide major improvement over Michel et al. (2023), which achieves $O(K/\Delta_S + K/(\Delta_S^*)^2)$ satisficing regret. Compared to Michel et al. (2023), our regret bound remove the dependence on $\Delta_S$ and the additional $1/\Delta_S^*$ factor. We also point out that if we want to achieve the same satisficing regret as Garivier et al. (2019), we can simply change the LCB test in Step 3 to a UCB test.*

**2. Concave Bandits:** Consider a concave bandit problem, *i.e.*, $\mathcal{X} \subset \mathbb{R}^d$ is a bounded convex set, and $r(X)$ is a concave and 1-Lipschitz continuous function defined on $\mathcal{X}$. Agarwal et al. (2011) gives an algorithm that enjoys $\text{poly}(d)\sqrt{T} \cdot \text{polylog}(T)$ regret. Together with Theorems 1 and 2, we have the following corollary.

**Corollary 2.** *By using the algorithm in Agarwal et al. (2011) as* ALG*, if* $r(X^*) \geq S$*, then the satisficing regret of* SELECT *is bounded by*

$$Regret_S \leq \min \left\{ \frac{\mathrm{poly}(d)}{\Delta_S^*} \cdot \mathrm{polylog}\left(\frac{d}{\Delta_S^*}\right), \mathrm{poly}(d)\sqrt{T} \cdot \mathrm{polylog}(T) \right\};$$

*If* $r(X^*) < S$*, the standard regret of* SELECT *is bounded by* $Regret \leq \mathrm{poly}(d)\sqrt{T} \cdot \mathrm{polylog}(T)$.

**3. Lipschitz Bandits:** Consider a Lipschitz bandit problem in $d$ dimensions, *i.e.*, $\mathcal{X} = [0,1]^d$, and $r(X)$ is an $L$-Lipschitz function (here Lipschitz functions are defined in the sense of $\infty$-norm). Bubeck et al. (2011) introduces a uniformly discretized UCB algorithm to achieve a $O(L^{d/(d+2)}T^{(d+1)/(d+2)}\sqrt{\log(T)})$ regret upper bound bound. Together with Theorem 1 and Theorem 2, we have the following corollary.

**Corollary 3.** *By using the UCB algorithm with uniform discretization in Bubeck et al. (2011) as* ALG*, if* $r(X^*) \geq S$*, then the satisficing regret of* SELECT *is bounded by*

$$Regret_S \leq \min \left\{ \frac{L^d}{(\Delta_S^*/2)^{d+1}} \cdot \mathrm{polylog}\left(\frac{L}{\Delta_S^*}\right), L^{\frac{d}{d+2}}T^{\frac{d+1}{d+2}} \cdot \mathrm{polylog}(T) \right\};$$

*If* $r(X^*) < S$*, the standard regret of* SELECT *is bounded by* $Regret \leq L^{\frac{d}{d+2}}T^{\frac{d+1}{d+2}} \cdot \mathrm{polylog}(T)$.

**Remark 4.** *Recall that* $\Delta_S = \min\{S - r(X) : r(X) < S\}$*, one can easily verify that in the realizable case,* $\Delta_S = 0$ *for both concave bandits and the Lipschitz bandits. As such, one cannot directly apply the results in Garivier et al. (2019); Michel et al. (2023) to acquire a constant satisficing regret. With the notion of exceeding gap* $\Delta_S^*$ *and* SELECT*, we establish constant satisficing regret bounds for these two settings.*

**4. Lower Bounds:** To complement our main results, we also present the satisficing regret lower bounds for finite-armed bandits and concave bandits. The first one is a satisficing regret lower bound for the finite-armed bandits.

**Theorem 3** (Finite-Armed Bandits)**.** *For every non-anticipatory learning algorithm* $\pi$*, every* $\Delta > 0$ *and* $T \geq 1/\Delta^2$*, there exists an instance of two-armed bandit such that* $r(X^*) - S = \Delta$*, and the satisficing regret incurred by* $\pi$ *is at least* $\Omega(1/\Delta)$.

**Remark 5.** *In Michel et al. (2023), the authors adapt the results from Bubeck et al. (2013) (for MAB with known optimal mean reward) to establish a* $\Omega(1/\Delta_S)$ *satisficing regret lower bound for finite-armed bandits. This is different than the one in Theorem3, which is* $\Omega(1/\Delta)$ *with* $\Delta$ *being the exceeding gap (i.e.,* $\Delta_S^*$*). This difference originates from the lower bound instances. The lower bound instance in Bubeck et al. (2013) (when adapted to satisficing bandits) sets the threshold value to the optimal mean reward, i.e.,* $S = r(X^*)$*, which makes both the satisficing regret and standard regret lower bounds to be* $\Omega(1/\Delta_S)$*. In our lower bound proof, we allow* $S$ *to be smaller than* $r(X^*)$*, which leads to a lower bound that depends on the difference between these two quantities.*

Next, we provide a satisficing regret lower bound for bandits with concave reward.

**Theorem 4** (Bandits with Concave reward)**.** *For every non-anticipatory learning algorithm* $\pi$*, every* $\Delta > 0$ *and every* $T \geq 1/\Delta^2$*, there exists an instance of* 1*-dimensional bandit with concave reward such that* $r(X^*) - S = \Delta$*, and the satisficing regret incurred by* $\pi$ *is at least* $\Omega(1/\Delta)$.

## 6 NUMERICAL EXPERIMENTS

In this section, we conduct numerical experiments to test the performance of SELECT on finite-armed bandits, concave bandits, and Lipschitz bandits. All experiments are run on a PC with 4-core CPU and 16GB of RAM.

### 6.1 FINITE-ARMED BANDITS

**Setup:** In this case, we consider an instance of $4$ arms, and the expected reward of all arms are $\{0.6, 0.7, 0.8, 1\}$. The length of the time horizon is set to be $500$ to $5000$ with a stepsize of $500$. We

conduct experiments for both the realizable case and the non-realizable case. For the realizable case, we set the satisficing level $S = 0.93$; for the non-realizable case, we set the satisficing level $S = 1.5$. In both cases, we compare SELECT against Thompson sampling in Agrawal & Goyal (2017) as well as SAT-UCB and SAT-UCB+ in Michel et al. (2023). For SELECT, we use Thompson Sampling as the learning oracle. The experiment is repeated for 1000 times and we report the average results.

**Results:** The results of the realizable case are presented in Figure 2(a), and the results of the non-realizable case are presented in Figure 2(b). From Figure 2(a) one can see that SELECT, SAT-UCB and SAT-UCB+ exhibit constant satisficing regret, and incur less satisficing regret compared to Thompson sampling. Furthermore, SELECT incurs less regret compared to SAT-UCB and achieves comparable performance with SAT-UCB+ in the realizable case. From Figure 2(b) one can see that SELECT incurs roughly the same standard regret as Thompson sampling. Furthermore, SELECT also incurs less standard regret in the non-realizable case compared to either SAT-UCB or SAT-UCB+.

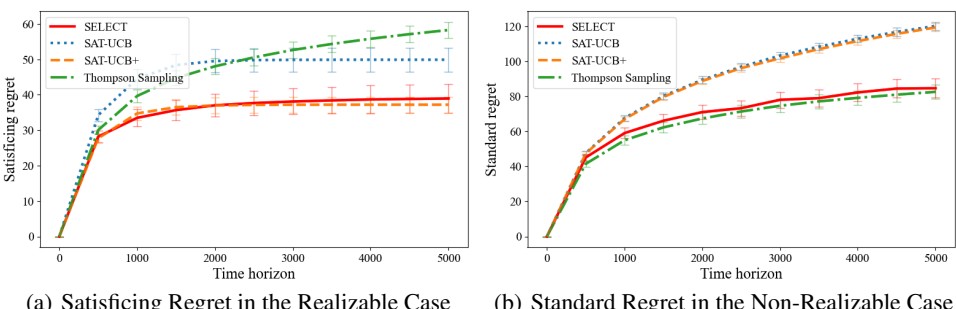

(a) Satisficing Regret in the Realizable Case  (b) Standard Regret in the Non-Realizable Case

Figure 2: Comparison of SELECT , Thompson sampling, SAT-UCB and SAT-UCB+

## 6.2 CONCAVE BANDITS

**Setup:** In this case, we consider an instance with arm set $[0, 1]$ and concave reward function $r(x) = 1 - 16(x - 0.25)^2$. The length of the time horizon is set to be 500 to 5000 with a stepsize of 500. We consider both the realizable case and the non-realizable case. For the realizable case, we set the satisficing level $S = 0.3$; for the non-realizable case, we set the satisficing level $S = 1.5$. For both cases, we compare our algorithm SELECT with the convex bandit algorithm introduced in Agarwal et al. (2011). For SELECT, we use the algorithm in Agarwal et al. (2011) as the learning oracle. We also use SAT-UCB and SAT-UCB+ as heuristics by viewing the problem as Lipschitz bandits. That is, we first discretize the arm space uniformly with stepsize $L^{-1/3}T^{-1/3}$, where $L$ is the Lipschitz constant of the reward function, and then run SAT-UCB and SAT-UCB+ over the discretized arm space. The experiment is repeated for 1000 times and we report the average results.

**Results:** The results of realizable case are provided in Figure 3(a), and the results of non-realizable case are provided in Figure 3(b). From Figure 3(a), we can see that SELECT converges to satisficing arms faster and incurs smaller satisficing regret than the algorithm in Agarwal et al. (2011), SAT-UCB and SAT-UCB+. The reason why SAT-UCB and SAT-UCB+ has not converged is that they usually repeatedly pull non-satisficing arms which are close to satisficing level but provide empirically highest UCB. As shown in Figure 3(b), SELECT can also reach good enough distribution compared to the algorithm in Agarwal et al. (2011) , SAT-UCB, and SAT-UCB+ in the non-realizable case.

## 6.3 LIPSCHITZ BANDITS

**Setup:** In this case, we consider a two-dimensional Lipschitz bandit in domain $(x, y) \in [0, 1]^2$ with Lipschitz reward $f(x, y) = \min\{1, 3e^{-100((x-0.5)^2+(y-0.7)^2)}\}$. The length of the time horizon is set to be 500 to 5000 with a stepsize of 500. We consider both the realizable case and the non-realizable case. For the realizable case, we set the satisficing level S = 0.5; for the non-realizable case, we set the satisficing level S = 1.5. For both cases, we compare our algorithm SELECT with the uniformly discretized UCB introduced in Bubeck et al. (2011) (hereafter referred to as the "Uniform UCB").

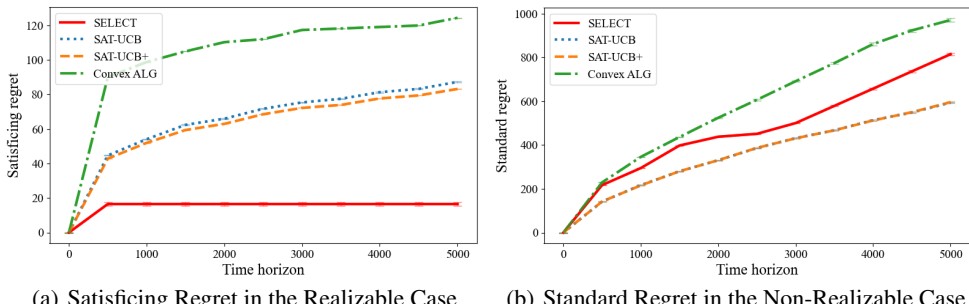

(a) Satisficing Regret in the Realizable Case     (b) Standard Regret in the Non-Realizable Case

Figure 3: Comparison of `SELECT` , Stochastic Convex, SAT-UCB and SAT-UCB+ for concave bandit

For `SELECT`, we use the Uniform UCB as the learning oracle. We again use SAT-UCB and SAT-UCB+ as heuristics by uniformly discretizing the arm space with stepsize $L^{-1/4}T^{-1/4}$, where $L$ is the Lipschitz constant of the reward function, and run the algorithms over the discretized arms. The experiment is repeated for 1000 times and we report the average results.

**Results:** The results of realizable case are provided in Figure 4(a), and the results of non-realizable case are provided in Figure 4(b). From Figure 4(a), we can see that `SELECT` converges to satisficing arms faster and incurs smaller satisficing regret than Uniform UCB, SAT-UCB and SAT-UCB+. From 4(b), we can see that `SELECT` still works the best while Uniform UCB, SAT-UCB and SAT-UCB+ have similar performance.

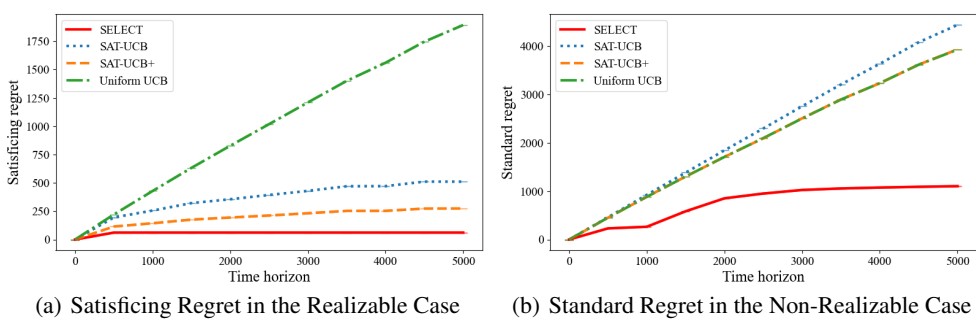

(a) Satisficing Regret in the Realizable Case     (b) Standard Regret in the Non-Realizable Case

Figure 4: Comparison of `SELECT` , Uniform UCB, SAT-UCB and SAT-UCB+ for Lipschitz bandit

## 7 CONCLUSION

In this paper, we propose `SELECT`, a general algorithmic template, for satisficing regret minimization in bandit optimization. For a given bandit optimization problem and a sub-linear regret learning oracle, we show that `SELECT` attains constant satisficing regret in the realizable case and the same regret as the learning oracle in the non-realizable case. We instantiate `SELECT` to finite-armed bandits, concave bandits, and Lipschitz bandits, and also make a discussion on the corresponding lower bounds. Finally, numerical experiments show that `SELECT` attains constant regret in several popular bandit problems and demonstrates excellent performances when compared to other benchmarks in both the realizable and the non-realizable cases.

For future studies, we suggest further exploration in the following directions. First, our work mainly focuses on the stationary settings, and further research could explore satisficing in the nonstationary online learning settings (Wei & Luo, 2021; Cheung et al., 2022). Also, the notion of satisficing regret indeed discourages exploration. Therefore, another direction is to study the role of satisficing regret in settings with limited adaptivity (Gao et al., 2019; Feng et al., 2023).

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
