# OpenReview forum: "Satisficing Regret Minimization in Bandits"
_ICLR.cc/2025/Conference — ICLR 2025 Poster_

### Official Review · Reviewer_EjZr · 2024-10-29

**Soundness:** 3
**Presentation:** 3
**Contribution:** 2
**Rating:** 6
**Confidence:** 4

**Summary:**

This paper addresses the problem of satisficing exploration, inspired by the concept of satisficing in decision-making. The authors propose an algorithmic framework called SELECT, which leverages a learning oracle with sub-linear regret guarantees to iteratively identify and test potential satisficing arms. SELECT achieves constant regret in the realizable cases, and it maintains the original regret bound in the non-realizable cases. Finally, numerical experiments are conducted to demonstrate the algorithm's performance across various bandit settings.

**Strengths:**

1. It is commendable that the proposed algorithm can be applied to any learning oracle with sub-linear regret guarantees, making it adaptable to different bandit models.
2. The three-step algorithm design is well-constructed and clearly explained. Step 3 (LCB Tests) is particularly impressive. Roughly speaking, in the realizable cases, a certain round will continue indefinitely, while in the non-realizable cases, the algorithm will proceed round by round.

**Weaknesses:**

1. While the three-step design in each round is great, each round of SELECT runs independently, resembling the doubling trick, which is often criticized. In the non-realizable case, this may lead to suboptimal theoretical and practical performance.
2. The result that constant regret can be achieved in the realizable case is not surprising, given prior research like Garivier et al. (2019). Additionally, the study on the lower bound feels insufficient. For instance, in the case of finite-armed bandits, the lower bounds are limited to two-armed bandits. I encourage the authors to explore the lower bounds further.
3. A minor issue to note is the terminology "satisficing exploration." In the bandit literature, exploration refers to selecting arms with uncertain rewards to gather information, as opposed to exploitation, where arms are selected to maximize immediate rewards based on current knowledge. In this problem, there is indeed a tradeoff between exploration and exploitation. I believe the thresholding bandits problem (in the pure exploration setting) is a better model of satisficing exploration. The authors might consider clarifying this or adopting different terminology.

**Questions:**

1. For Condition 1, what happens when $\alpha < 1/2$?
2. Regarding the numerical results for finite-armed bandits, which algorithm is used as the learning oracle? Could you explain why SELECT outperforms Uniform UCB in Figure 4b?
3. Can $\gamma_i$ be multiplied by a constant? If so, are the empirical results sensitive to the choice of constant for both realizable and non-realizable cases?

---

> ### Author Response · Authors · 2024-11-20
> **Response for Reviewer EjZr (Part 1)**
>
> Thanks for the great effort in reviewing our work! We provide a detailed response below for your comments.
>
> **1. Each round running independently**
>
> Our algorithm indeed runs different rounds independently, but this design is mainly for the simplicity of regret analysis. In some special cases such as finite-armed bandits and linear bandits with discrete arm set, we can easily make use of historical data from previous rounds to improve empirical performance of our algorithm. For example, if the learning oracle is a UCB-based algorithm, then we can make use of data from previous rounds to construct confidence intervals. In LCB tests, we can also make use of historical data on the candidate satisficing arm to construct lower confidence bounds. We believe the same regret bounds still hold with the use of historical data.
>
> In terms of performance of our algorithm in the non-realizable case, we prove in Theorem 2 that our algorithm enjoys the same (up to logarithmic factors) standard regret bound as the learning oracle we use in step 1, under the mild condition of $\alpha\geq 1/2$. In particular, in finite-armed bandits, concave bandits and Lipschitz bandits, the standard regret bound of our algorithm in the non-realizable case is in fact near-optimal.
>
> Regarding the practical performance, Figures 2(b), 3(b), and 4(b) also demonstrate the robust empirical performance of our algorithm in non-realizable cases. As shown, in all three cases, our algorithm achieves competitive performance against the best benchmark.
>
> **2. Key contributions**
>
>
> We would like to highlight that existing algorithms such as the ones introduced in [1] and [2], whose satisficing regret bound depends on the minimum satisficing gap, are only limited to the setting of finite-armed bandits. These algorithms are not capable of handling more general classes of bandit problems such as concave bandits, Lipschitz bandits or linear bandits, where the minimum satisficing gap can simply be $0$ (see lines 68 - 75 for a detailed discussion). In fact, as shown in Figures 3(a) and 4(a), the adapted versions of SAT-UCB and SAT-UCB+ (initially proposed in [2] for finite-armed bandits) are not able to achieve constant satisficing regret in the concave and Lipschitz bandit settings. As such, it is not immediately clear if constant satisficing regret is possible beyond finite-armed bandits.
>
> To this end, our work serves to provide the first affirmative answer. More specifically, we establish a general approach that attains constant satisficing regret for a wide range of prevalent bandit settings (e.g., those demonstrated in Section 5) under the realizable case (see lines 90 - 96 for a more detailed discussion). Moreover, our approach also attains competitive performance in the non-realizable case.
>
>
>
> **3. Lower bound**
>
> Our lower bound construction is built on the results in Theorem 6 in [4], which focuses on two-armed bandits. We believe one way to go beyond two-armed bandits is to follow the results in Theorem 10 in [5], which shares a similar flavor to Theorem 6 in [4]. We will examine this and include any improved results in our final version.
>
> **4. Terminology "Satisficing Exploration"**
>
> Thank you for pointing this out. We change the title and the name of the algorithm to reflect this point in our revised paper.
>
> **5. $\alpha<1/2$**
>
> Thank you for this great question. In our proof for Theorem 1, we prove that in round $i$, the satisficing regret incurred by step 1 is bounded by $\gamma_i^{-\alpha/(1-\alpha)}\cdot\text{polylog}(\gamma_i)$, while the satisficing regret incurred by step 2 is bounded by $\gamma_i^{-1}\cdot\text{polylog}(\gamma_i)$ (here we omit dependence on the problem class specific constant $C_1$). The reason why the current results rely on $\alpha\geq 1/2$ is that only when $\alpha\geq 1/2$, the satisficing regret incurred in step 1 dominates that incurred in step 2. If $\alpha<1/2$, then the satisficing regret incurred in step 2 dominates that incurred in step 1, and using the same analysis we are only able to obtain a $1/\Delta_S^*\cdot\text{polylog}(1/\Delta_S^*)$ satisficing regret bound instead of the satisficing regret bound stated in Theorem 1. Due to the same reason, if $\alpha<1/2$, we are only able to obtain a $\sqrt{T}\cdot\text{polylog}(T)$ standard regret bound instead of $T^\alpha\cdot\text{polylog}(T)$. On the other hand, we believe it is reasonable to assume $\alpha\geq 1/2$, because even in the simplest two-armed bandit setting we have $\alpha=1/2$, as the standard regret bound of UCB for two-armed bandits is $11\sqrt{2T\log(T)}$ (see Theorem 7.2 of [3]), and there is a worst-case regret lower bound of $1/27\sqrt{T}$ (see Theorem 15.2 of [3]).
>
> **6. Oracle used in for finite-armed bandits**
>
> In Figure 2(b), we use Thompson sampling as our learning oracle. We have clarified this in the revised paper (see line 439-440).

---

> ### Author Response · Authors · 2024-11-20
> **Response for Reviewer EjZr (Part 2)**
>
> **7. The reason why $\texttt{SELECT}$ outperforms Uniform UCB in Figure 4(b)**
>
> We explain the reason why $\texttt{SELECT}$ outperforms Uniform UCB as follows. Uniform UCB is an algorithm that achieves an optimal standard regret bound in the worst case for Lipschitz bandits, and particularly works well in worst-case instances (see [6] for a detailed discussion). However, in the instance we use in the experiment, both the set of optimal arms and the set of significantly suboptimal arms are large, which is very different from the worst-case instances of Lipschitz bandits. Therefore, in this particular instance, directly running Uniform UCB over the entire time horizon tend to over-explore (i.e., the number of discretized arms increases with longer time horizon) and incur significant regret. In contrast, $\texttt{SELECT}$ starts with running Uniform UCB over a small number of time steps $t_i$. This allows $\texttt{SELECT}$ to have a much smaller number of discretized arms and smaller confidence intervals in early stages of the algorithm, thus explores less and exploits more aggressively. Therefore, although in the non-realizable case, $\texttt{SELECT}$ may underperform Uniform UCB by a logarithmic factor of $T$ in the worst case, in the instance used in Figure 4(b), $\texttt{SELECT}$ outperforms Uniform UCB.
>
> **8. Robustness in the choice of $\gamma_i$**
>
> All our theoretical results still hold if all $\gamma_i$ are multiplied by a constant. We add a numerical experiment in Appendix C of our revised paper to test the robustness of $\texttt{SELECT}$ in the choice of $\gamma_i$. The results show that different choices of $\gamma_i$ have limited impact on the empirical performance of our algorithm in both realizable and non-realizable settings. Therefore, the empirical performance of $\texttt{SELECT}$ is robust in
> the choice of $\gamma_i$.
>
>
> [1] Garivier, A., Ménard, P., \& Stoltz, G. (2019). Explore first, exploit next: The true shape of regret in bandit problems. Mathematics of Operations Research, 44(2), 377-399.
>
> [2] Michel, T., Hajiabolhassan, H., \& Ortner, R. (2022). Regret bounds for satisficing in multi-armed bandit problems. Transactions on Machine Learning Research.
>
> [3] Lattimore, T., \& Szepesvári, C. (2020). Bandit algorithms. Cambridge University Press.
>
> [4] Bubeck, S., Perchet, V., \& Rigollet, P. (2013). Bounded regret in stochastic multi-armed bandits. In Conference on Learning Theory. PMLR, 122-134.
>
> [5] Lattimore, T. (2018). Refining the confidence level for optimistic bandit strategies. Journal of Machine Learning Research, 19(20), 1-32.
>
> [6] Bubeck, S., Stoltz, G., \& Yu, J. Y. (2011). Lipschitz bandits without the lipschitz constant. In Algorithmic Learning Theory: 22nd International Conference, ALT 2011, Espoo, Finland, October 5-7, 2011. Proceedings 22 (pp. 144-158). Springer Berlin Heidelberg.

---

> > ### Comment · Reviewer_EjZr · 2024-11-23
> >
> > Thank you for your response. Regarding the first point, using historical data from previous rounds can indeed often improve empirical performance. However, analyzing such algorithms is challenging because the interdependence between different rounds is difficult to address. Similar to the well-known doubling trick, order-wise optimality can be maintained by ignoring historical data. Additionally, the proposed algorithm seems to require more practical consideration regarding the selection of parameters and the base algorithm. Therefore, I will maintain my initial rating.

---

> > > ### Author Response · Authors · 2024-11-23
> > >
> > > Thank you again for the constructive feedback! We tend to agree with you that some of the points may worth further study and we shall suggest them as future work in the final version.

---

### Official Review · Reviewer_Y6qp · 2024-10-29

**Soundness:** 2
**Presentation:** 3
**Contribution:** 3
**Rating:** 6
**Confidence:** 4

**Summary:**

The paper introduces SELECT, an algorithmic framework designed for satisficing exploration in bandit optimization. The primary objective of SELECT is to frequently identify arms with mean rewards exceeding a specified threshold, with its performance evaluated through satisficing regret, which measures the cumulative deficit of the chosen arm's mean reward compared to this threshold. SELECT operates by leveraging a learning oracle that provides a sub-linear regret upper bound. It iteratively identifies potential satisficing arms, collects data samples, and monitors the LCB of the arm’s mean reward against the threshold to determine if it qualifies as a satisficing arm. The algorithm guarantees constant satisficing regret in scenarios where a satisficing arm exists (realizable case) and matches the standard regret of the oracle in non-realizable situations. The framework is successfully instantiated across various bandit settings. Numerical experiments validate the efficacy of SELECT, demonstrating its ability to achieve constant regret.

**Strengths:**

- The paper successfully integrates standard bandit optimization algorithms into a new framework, achieving better results and providing a more direct application of bandit algorithms to the satisficing exploration problem.
- The proposed SELECT employs a unique approach by utilizing a learning oracle for bandit optimization, enabling it to sample candidate arms and monitor performance efficiently.
- The paper establishes that SELECT achieves a constant satisficing regret in realizable cases, independent of the satisficing gap, and inversely related to the exceeding gap. This feature allows it to maintain performance even when the satisficing gap varies.

**Weaknesses:**

- The main weakness is that the algorithm imposes stringent conditions (Condition 1) on the oracle algorithm, requiring sublinear regret for all time steps t. Most algorithms, including all oracles referenced in Section 5, only achieve sublinear regret when t is sufficiently large. If alpha approaches 1 when t is small, the theoretical regret bound could become excessively large.
- The algorithm involves hyperparameters that rely on the oracle algorithms. In scenarios where oracles are unavailable or when the sublinear oracles have unclear parameters, extending theoretical conclusions becomes challenging.
- The first step of each phase necessitates a bandit algorithm, which is crucial. However, the paper lacks a general discussion on how to select the appropriate oracle algorithm.
- While Remark 2 highlights the novelty of each step, an ablation study demonstrating the impact of each component would strengthen the paper. Currently, the experimental results do not robustly support the conclusions, as SELECT only outperforms all baselines in 3 out of 6 settings.

**Questions:**

- There appears to be an inconsistency in the paper regarding the baseline references to "Hajiabolhassan \& Ortner (2023)" in line 427 and "Michel et al. (2023)". Clarification is needed.
- The time horizon T may not be large enough for UCB-based algorithms. For instance, in Figure 3(b), as T further increases,  SELECT appears to perform worse than the other algorithms.
- The experimental results further indicate that Condition 1 is overly strict, as the oracles struggle to satisfy it. For example, in Figures 4(a) and 4(b), the regret appears to converge to T for Uniform UCB.

---

> ### Author Response · Authors · 2024-11-20
> **Response for Reviewer Y6qp (Part 1)**
>
> Thank you so much for providing these constructive comments! We provide a detailed response below to your comments.
>
> **1. Condition 1**
>
> We double-check Condition 1 and we believe it can be satisfied by many bandit problems in the literature as long as $t\geq 2$ (this is to avoid the corner case of $\log(t)=0$ when $t=1$), including those demonstrated in Section 5.
> In what follows, we revisit the algorithms used in Section 5. Specifically, for every one of them, we provide the **exact choices of $C_1,\alpha,$ and $\beta$ under which Condition 1 is satisfied:**
>
> **(a) Finite-armed bandits:** Consider any instance of finite-armed bandits with mean reward bounded by $[0,1]$. By Theorem 7.2 of [1], for any given time horizon $t\geq 2$, the standard regret of the UCB algorithm is upper bounded by $8\sqrt{Kt\log(t)}+3\sum_{k=1}^K (r(k^*)-r(k))\leq 8\sqrt{Kt\log(t)}+3K$, where $K\geq 2$ is the number of arms and $k^*$ is the optimal arm. When $t>K$, the standard regret is upper bounded by $8\sqrt{Kt\log(t)}+3K\leq 11\sqrt{Kt\log(t)}$, where the last inequality is due to $t>K\geq 2$, thus $t\geq 3$ and $\log(t)\geq \log(3)>1$; When $t\leq K$, the standard regret is upper bounded by $t\leq \sqrt{Kt}\leq 11\sqrt{Kt\log(2)}\leq 11\sqrt{Kt\log(t)}$. Therefore, for any given time horizon $t\geq 2$, the regret of the UCB algorithm is upper bounded by $11\sqrt{Kt\log(t)}$. By setting $C_1=11\sqrt{K}$ and $\alpha=\beta=1/2$ and using the UCB algorithm as the learning oracle, we verify that finite-armed bandits satisfy Condition 1.
>
> **(b) Concave bandits:** We use the special case of one-dimensional concave bandits as an example. Consider any one-dimensional concave bandits with arm set being $[0,1]$ and reward function being $1$-Lipschitz. By Theorem 1 of [2], for any given time horizon $t\geq 2$, the standard regret of Algorithm 1 in [2] is upper bounded by $108\sqrt{t\log(t)}\cdot\log_{4/3}(t)=108/\log(4/3)\cdot\sqrt{t}\cdot(\log(t))^{3/2}$. Therefore, by setting $C_1=108/\log(4/3)$, $\alpha=1/2$, $\beta=3/2$ and using Algorithm 1 in [2] as the learning oracle, we verify that one-dimensional concave bandits satisfy Condition 1. In fact, similar arguments also hold for concave bandits in higher dimensions (see Algorithm 2 and Theorem 2 of [2], we also added a detailed discussion on this in Appendix A.2 of the revised paper).
>
> **(c) Lipschitz bandits:** Consider any Lipschitz bandits with arm set being $[0,1]^d$, Lipschitz constant $L$ and mean reward bounded by $[0,1]$ (here we define Lipschitz continuity in the sense of $\infty$-norm). By Section 2.2 of [3], for any given time horizon $t\geq 2$, the standard regret of the uniformly discretized UCB algorithm introduced in [3] (hereafter referred to as the ``Uniform UCB") is upper bounded by $(1+c_{\text{ALG}})L^{d/(d+2)}t^{(d+1)/(d+2)}\cdot\sqrt{\log(t)}$ if the regret of the UCB algorithm for finite-armed bandits with $K$ arms is upper bounded by $c_{\text{ALG}}\sqrt{Kt\log(t)}$ for any $t\geq 2$. Using our discussions in (a) we have $c_{\text{ALG}}=11$, the regret of the Uniform UCB algorithm is bounded by $12L^{d/(d+2)}t^{(d+1)/(d+2)}\cdot\sqrt{\log(t)}$. Therefore, by setting $C_1=12L^{d/(d+2)}$, $\alpha=(d+1)/(d+2)$, $\beta=1/2$ and using Uniform UCB as the learning oracle, we verify that Lipschitz bandits satisfy Condition 1.
>
> In summary, in all three examples above, the regret of the oracle algorithm has an upper bound in the form of $C_1t^\alpha\cdot(\log(t))^\beta$ **for all given time horizon** $t\geq 2$. Notably, the value of $\alpha$ **remains unchanged for all $t\geq2$**. That is to say, $\alpha$ will not approach $1$ even if $t$ is small, and the theoretical satisficing regret bound holds for all $t\geq 2$. We have also added a detailed discussion on this in Appendix A of the revised paper.
>
> **2. Hyperparameters in the algorithm**
>
> We note that the only hyperparameter used in our algorithm is $\alpha$. In most of the bandit problems considered in literature, the exact value of $\alpha$ is known (see, e.g., examples provided in the previous point). If the exact $\alpha$ is unknown but an upper bound of $\alpha$ strictly smaller than $1$ is available, we can also plug in the upper bound of $\alpha$ when running our algorithm.

---

> ### Author Response · Authors · 2024-11-20
> **Response for Reviewer Y6qp (Part 2)**
>
> **3. Reliance on a learning oracle and learnability**
>
> We would like to highlight that the main contribution of our paper is to provide a general approach to achieve constant satisficing regret bound in the realizable case **whenever a sub-linear standard regret learning oracle exists** for the corresponding problem class (see lines 19-23 in the Abstract). This condition (i.e., a sub-linear standard regret learning oracle exists) is satisfied by many popular bandit problem classes, including those considered in Section 5 and linear bandits.
>
> Indeed, our approach cannot handle problem classes without a sub-linear standard regret learning oracle, but this goes beyond the scope of our work. We reiterate that, as pointed out in lines 68 - 75, prior approaches for satisficing bandits can only handle finite-armed bandits, and it is not even clear if constant satisficing regret is possible beyond finite-armed bandits. Therefore, our work mainly focuses on answering this question and we establish a general approach that attains constant satisficing regret for a wide range of prevalent bandit settings, e.g., those demonstrated in Section 5.
>
> **4. Selection of learning oracle**
>
> In general, given a bandit problem class, any learning oracle with a sub-linear standard regret bound in the problem class can be used in step 1 of our algorithm to obtain a constant satisficing regret bound. For example, for finite-armed bandits, we use the UCB algorithm in step 1; for concave bandits, we use Algorithm 2 in [2] in step 1; for Lipschitz bandits, we use Uniform UCB in Section 2.2 of [3] in step 1 (see Section 5 of the paper and our first point).
>
> **5. Ablation study**
>
> Following your suggestion, we added an ablation study in Appendix B of the revised paper to show that all three steps are necessary for $\texttt{SELECT}$ in obtaining a constant satisficing regret bound. Specifically, we use an instance of Lipschitz bandit, and compare the performance of $\texttt{SELECT}$ with the following adapted versions of $\texttt{SELECT}$ where one of the three steps is removed:
>
> **(a) $\texttt{SELECT}$ without step 1:** Instead of identifying candidate satisficing arms using a learning oracle, the candidate satisficing arm is drawn uniformly at random from the arm set before we proceed to forced sampling and LCB tests in each round;
>
> **(b) $\texttt{SELECT}$ without step 2:** After identifying a candidate satisficing arm from step 1 in each round, the forced sampling in step 2 is skipped and a LCB test on the candidate satisficing arm is immediately started;
>
> **(c) $\texttt{SELECT}$ without step 3:** Each round is terminated after running step 1 and step 2 without entering the LCB test in step 3.
>
> In the ablation study, we show that if step 1 is removed, then the algorithm struggles to identify satisficing arms, incurring almost linear in $T$ satisficing regret. If either step 2 or step 3 is removed, each round is almost always terminated shortly after completing step 1, thus the algorithm is unable to maintain constant satisficing regret because of frequent reruns of the learning oracle. Therefore all three steps play a vital role in obtaining a constant satisficing regret bound in the realizable case, and the algorithm fails to maintain a constant satisficing regret bound if any of the three steps is removed. We also refer to Remark 2 (line 251-278) for a detailed discussion on the purpose of each component of $\texttt{SELECT}$.
>
> **6. Empirical performance of our algorithm**
>
> We would like to emphasize that the algorithm designed in our work focuses mainly on achieving constant satisficing regret bound in the realizable case while still providing robust performance in the non-realizable case **for a wide range of prevalent bandit settings**. That being said, some empirical performance tradeoff is perhaps unavoidable. Nevertheless, our algorithm still renders competitive performance in the non-relizable cases.
>
> In all three realizable settings, the satisficing regret of our algorithm clearly converges to a constant as time horizon $T$ increases, which matches our theoretical results. In the finite-armed bandit setting, i.e., Figure 2(a), our algorithm slightly underperforms SAT-UCB+ by around $5$ percent but outperforms all other benchmarks. Given that SAT-UCB+ is specifically designed heuristic (with no theoretical performance guarantee) for finite-armed bandits and our algorithm is a general framework that handles a much more general class of bandit problems, it is not surprising that SAT-UCB+ will slightly outperform our algorithm in the finite-armed bandit setting. In both Lipschitz bandits and concave bandits, i.e., Figure 3(a) and Figure 4(a), our algorithm outperforms all of the three benchmarks. Furthermore, our algorithm is able to demonstrate constant satisficing regret while none of the benchmarks are able to achieve constant satisficing regret.

---

> ### Author Response · Authors · 2024-11-20
> **Response for Reviewer Y6qp (Part 3)**
>
> **7. Reference in line 427**
>
> Thank you for catching this. The correct reference should be Michel et al. (2023), and we have fixed it in the revised paper.
>
> **8. Performance of $\texttt{SELECT}$ in Figure 3(b)**
>
> We have increased the maximum time horizon to $50000$. The results of concave bandits in the non-realizable case with maximum time horizon $50000$ is provided in Figure 7 of Appendix G of the revised paper. One can observe from Figure 7 that the regret of $\texttt{SELECT}$ remains sub-linear in $T$ under the increased time horizon. In fact, the empirical performance of $\texttt{SELECT}$ is consistently better than Algorithm 1 in [2] even after we increase the maximum time horizon to $50000$.
>
> As shown in Figure 7, the standard regret of $\texttt{SELECT}$ in the non-realizable case exhibits a wave shape. This is mainly because the algorithm runs in rounds. At the beginning of each round, the algorithm starts a rerun of the learning oracle, which could result in a rapid increase in standard regret in the early stage of each round. In later stages of every round, the learning oracle gradually converges to the optimal arm, and both step 2 and step 3 are exploiting a near-optimal arm obtained from the learning oracle, thus increase in standard regret slows down in later stages of each round. Similar phenomenon can also be observed in Figure 4(b).
>
> **9. Regret of Uniform UCB in Figure 4(a) and 4(b)**
>
> We have shown theoretically that Uniform UCB satisfies Condition 1 (see Appendix A.3 of the revised paper and the first point of the response). The performance of Uniform UCB shown in Figure 4 indeed aligns with the theoretical guarantee. As shown in the first point of our response, the regret of Uniform UCB is bounded by $12L^{1/2}T^{3/4}(\log(T))^{1/2}$. In our example, we have $L\approx 72$, and for example, when $T=5000$ the theoretical standard regret upper bound is $12L^{1/2}T^{3/4}\log(T)^{1/2}\approx 176690$. A comparison between the standard regret bound observed in Figure 4(b) and $0.015$ times the theoretical standard regret upper bound is provided in Figure 8 of Appendix G of the revised paper. From Figure 8 one can see that as $T$ increases, the standard regret observed in the experiment shares a similar growth pattern as the theoretical standard regret upper bound.
>
>
> [1] Lattimore, T., \& Szepesvári, C. (2020). Bandit algorithms. Cambridge University Press.
>
> [2] Agarwal, A., Foster, D. P., Hsu, D. J., Kakade, S. M., \& Rakhlin, A. (2011). Stochastic convex optimization with bandit feedback. Advances in Neural Information Processing Systems, 24.
>
> [3] Bubeck, S., Stoltz, G., \& Yu, J. Y. (2011). Lipschitz bandits without the lipschitz constant. In Algorithmic Learning Theory: 22nd International Conference, ALT 2011, Espoo, Finland, October 5-7, 2011. Proceedings 22 (pp. 144-158). Springer Berlin Heidelberg.

---

> > ### Comment · Reviewer_Y6qp · 2024-11-23
> >
> > I thank the authors for the response, which helped clarify most of my concerns. I've increased the score.

---

> > > ### Author Response · Authors · 2024-11-23
> > >
> > > Thank you very much for the positive feedback! We sincerely appreciate your efforts in reviewing our work.

---

### Official Review · Reviewer_pJ9Q · 2024-11-04

**Soundness:** 4
**Presentation:** 4
**Contribution:** 4
**Rating:** 8
**Confidence:** 4

**Summary:**

The paper tackles the problem of satisficing exploration in multi-armed bandits. Satisficing problem here represents finding an option which is above a preset threshold. The paper proposes a novel method SELECT which utilizes any existing bandit method with sub-linear regret guarantees and utilizes the same sample path trajectories to further provide a constant satisficing regret framework.

The method implementation is split into three parts:
1. Shadowing the sub-linear regret method's trajectory for a set number of rounds
2. Forced sampling of selected arm
3. Comparing the lower confidence bound of the selected arm with the threshold value.

The paper provides regret guarantees based on the difference between the highest mean among all the options and the threshold value (denoted in the paper as $\Delta_S*$). The paper also provides a matching lower bound (up-to-logarithmic factors) to validate the performance of SELECT.

The paper further goes on to provide examples of how SELECT can be used with different bandit frameworks, making the method quite applicable to a large set of setups. This is supplemented with experiments for the same, further strengthening the case of SELECT.

**Strengths:**

The following would contribute to the strengths of the paper:
- **Clear Writing**: The paper is well written, precise, and to-the-point.

- **Justified Problem Setup**: The paper clearly explains the justification of the problem setup, literature surround it and solution of the problem with theoretical and experimental backing.

- **Innovative Umbrella Solutions**: The novel proposed method SELECT can be appended to any sub-linear regret method and can provide constant satisfying performance guarantee to the respective application. This makes the algorithm quite applicable to a lot of varied problem setup.

- **Theoretical performance guarantees**: The paper provides theoretical proof on both the regret upper-bound and shows the tightness to the fundamental lower-bound on the best performance possible on the satisficing problem.

- **Example distinct setups**: The paper provides example problem setup in finite-armed bandits, concave bandits and Lipschitz bandits.

- **Experiments**: The paper provides a synthetic implementation for all the example setups and showcases the promise of SELECT method.

**Weaknesses:**

There are very few obvious loopholes to the paper. Overall the paper is a complete work. A few paragraphs on the potential future works and possible extensions would be a good addition.

**Questions:**

Nothing to add here

---

> ### Author Response · Authors · 2024-11-20
>
> Thanks for the great effort in reviewing our work! We added a paragraph in Conclusion (Section 7) in our revised paper and discussed some potential directions for future research (see line 537-539).

---

### Meta-Review · Area_Chair_rscs · 2024-12-21

**Metareview:**

This paper considers a variant of the bandit problem focused on exploring actions, referred to as *satisficing arms*, whose expected rewards exceed a given threshold. The authors propose an algorithm and provide a regret analysis. Weaknesses include the strong assumptions regarding the oracle algorithm, concerns about practical performance due to the use of doubling, and the regret lower bound being limited to the two-armed case. However, the authors have offered convincing responses to these review comments and have expressed a willingness to revise the paper. Given that the reviewers have reached a general consensus with positive opinions, I support the acceptance of this paper.

**Additional Comments On Reviewer Discussion:**

The reviewers expressed concerns about the strong assumptions regarding the oracle algorithm, the practical performance implications of using doubling, and the regret lower bound being restricted to the two-armed case. In response to these comments, the authors provided convincing answers and demonstrated a willingness to revise the paper.

---

### Decision · Program_Chairs · 2025-01-22

Accept (Poster)